# Using cellular device location data to estimate visitation to public lands: Comparing device location data to U.S. National Park Service's visitor use statistics

Wei-Lun Tsai[1]*, Nathaniel H. Merrill[2], Anne C. Neale[1], Madeline Grupper[3]

**1** United States Environmental Protection Agency, Office of Research and Development, Center for Public Health and Environmental Assessment, Public Health and Environmental Systems Division, Research Triangle Park, North Carolina, United States of America, **2** United States Environmental Protection Agency, Office of Research and Development, Center for Environmental Measurement and Modeling, Atlantic Coastal Environmental Sciences Division, Narragansett, Rhode Island, United States of America, **3** Oak Ridge Institute for Science and Education (ORISE) Research Fellow, Office of Research and Development, Center for Public Health and Environmental Assessment, Public Health and Environmental Systems Division, Research Triangle Park, North Carolina, United States of America

* tsai.wei-lun@epa.gov

⬆ OPEN ACCESS

**Data Availability Statement:** The data underlying the results presented in the study are available from https://doi.org/10.23719/1528093.

## Abstract

Understanding human use of public lands is essential for management of natural and cultural resources. However, compiling consistently reliable visitation data across large spatial and temporal scales and across different land managing entities is challenging. Cellular device locations have been demonstrated as a source to map human activity patterns and may offer a viable solution to overcome some of the challenges that traditional on-the-ground visitation counts face on public lands. Yet, large-scale applicability of human mobility data derived from cell phone device locations for estimating visitation counts to public lands remains unclear. This study aims to address this knowledge gap by examining the efficacy and limitations of using commercially available cellular data to estimate visitation to public lands. We used the United States' National Park Service's (NPS) 2018 and 2019 monthly visitor use counts as a ground-truth and developed visitation models using cellular device location-derived monthly visitor counts as a predictor variable. Other covariates, including park unit type, porousness, and park setting (i.e., urban vs. non-urban, iconic vs. local), were included in the model to examine the impact of park attributes on the relationship between NPS and cell phone-derived counts. We applied Pearson's correlation and generalized linear mixed model with adjustment of month and accounting for potential clustering by the individual park units to evaluate the reliability of using cell data to estimate visitation counts. Of the 38 parks in our study, 20 parks had a correlation of greater than 0.8 between monthly NPS and cell data counts and 8 parks had a correlation of less than 0.5. Regression modeling showed that the cell data could explain a great amount of the variability (conditional R-squared = 0.96) of NPS counts. However, these relationships varied across parks, with better associations generally observed for iconic parks. While our study increased our confidence in using cell phone data to estimate visitation, we also became aware of some of the limitations and challenges which we present in the Discussion.

**Funding:** The author(s) received no specific funding for this work.

**Competing interests:** The authors have declared that no competing interests exist.

## Introduction

Visiting natural areas contributes to people's health and wellbeing [1–3]. The availability, access and quality of natural spaces are important to people [4–6]. As such, quantifying and understanding the use of these resources informs efforts to protect and improve environmental quality and recreational opportunities. Accurate estimates of visitation across large geographic scales, management entities (e.g., national parks, national forests, state and local parks), and time are critical to park managers and surrounding communities to inform park planning efforts, including protecting resources and improving visitor safety [7], understanding environmental and economic benefits [8–10], and projecting how visitation may change over time based on changing environmental conditions such as climate effects [11]. However, quantifying the level of use and profile of the visitors of a resource is surprisingly and persistently hard to obtain.

Traditionally, quantifying visitation to natural areas has been achieved through various methods tailored to geography, management needs and budget. These include entrance receipts, traffic and trail counters, spot counts and aerial imagery [12, 13]. This is challenging, however, because protected and natural areas vary considerably in size, number of unmonitored entry points, proximity to urban areas, and other attributes. The range of characteristics of recreation units in the National Park System lead to the use of a variety of visitor counting methods [14]. At National Park units in the United States (U.S.), visitation counts are collected using traffic counters, trail counters, bus counts, on-site observations, ferry and aircraft counts, tickets, fixed-estimation (i.e., assumes a fixed number of visitors per day, week, or month), and estimation based on other locations within a park. These counting methods are often resource intensive. In addition, most of the parks use a combination of at least two methods. There are also variations within each method. For instance, parks that utilize traffic counters usually apply a constant for each vehicle (i.e., Person-Per-Vehicle, PPV), and this constant varies by park, location, and even season.

A recently-developed line of research aimed at scaling up the ability to estimate visitation across space and time is combining traditional methods with novel data sources and technology [15–18]. Social media have been used as one alternative source for estimating visitation [19]. Social media platforms, such as Flickr, Strava, Instagram, Twitter, and Facebook, often include geotagged locations along with posted photos or texts [16, 20–23]. These sources of information support potentially lower-cost and scalable methods of data collection.

Social media have been used to collect visitor use information in several studies. Fisher et al. [24] found that geotagged Flickr photos combined with trip reports on hiking forums correlated highly (ranging between 55% and 95%) with trail counter data in the National Forest System across the U.S. Similarly, Wilkins et al. [25] used data from geotagged Flickr photos taken in national parks to evaluate the effects of weather conditions on park recreation numbers. Social media data also offer an opportunity to collect a wealth and depth of information that would be hard to obtain without in-depth survey efforts, including visitors' spatial patterns and activities [26–29], values and perceptions [30], landscape preferences [20, 31], as well as equity of park access [32]. Despite the promise of using social media for visitor counts, there are still limitations. Social media data are collected passively, and therefore the availability of the data greatly depends not only on the popularity but also the policies of the platforms [16, 33]. A study [16] comparing the usability of three social media platforms (i.e., Instagram, Twitter, and Flickr) for visitor monitoring in parks in Finland and South Africa reported that social media data generally performed better in parks with more visitations, whereas social media data were found to be under- or over-estimated in less visited parks. In addition, the differences in the performance of the social media platforms were observed in these two countries;

Instagram had the best correlations with the official statistics in Finland, whereas Flickr performed the best in South Africa. There is also a risk of discontinued access to social media data, such as when Instagram terminated open access to their API and only offered access primarily for commercial purposes. Furthermore, the visitation data collected through social media can be biased by user demographics, which were found mostly to be women, aged 18 to 29, and residing in urban areas in most of the social media platforms [34, 35].

Cellular device location data (cell data hereafter) may offer a viable alternative that overcomes some of the challenges of social media data. Modern cell phones are equipped with Global Positioning Systems (GPS), which allows a device to record its location and share that location with smartphone applications. Cell data were widely used to understand human mobility patterns for informing public health actions during COVID-19 [e.g., 36, 37]. Before COVID-19, Fisher et al. [38] applied cell data to estimate visitation to outdoor recreation sites in South Korea and compared visitation using cell data and social media with visitation counts based on traditional survey methods. Visitor counts based on cell data were found to be comparable to the on-site counts [38]. Similar observations were found in later studies for a nature reserve in California [39] and for beaches in New England [40]. Cell data also have been applied to estimate change in recreation to coastal areas in response to changes in environmental conditions, such as bacterial-induced beach closures [41]. Like many other sources of big data (e.g., vehicle GPS, metro card, bank card), cell data can not only help understand human mobility patterns but also social, cultural, and economic values for better land management and city development [42, 43]. However, most of the existing studies using cell data to estimate visitation are applied in one site or sites with relatively similar geographic characteristics. Applying cell data across a large geographic area likely with great variations in settings (e.g., urbanicity, recognition) remains under explored.

The information coming from crowdsourced data on people's movement and choices around public lands provides new opportunities and new instruments to measure and understand the benefits of ecosystem services [19, 44–46]. The data also provide the ability for studying human behavior and perceptions of the environment, monitoring the environment, and supporting environmental planning and management [18]. While the opportunities are vast, examinations of the accuracy, feasibility, and limitations of applying these cell datasets across a wide range of types of public lands and for specific needs are prudent before these new data sources become integrated and accepted into management and policy analysis.

To understand the feasibility of cell data for quantifying visitation information and to address the research gap on the application of cell data across a large spatial scale, we examined the use of commercially available cell data for estimating visitation to public lands. We used the National Park Service's (NPS) visitor use statistics as a 'ground-truth' for the cell data and we provide an example application of the scalability of visitation models based on cell data to many parks across the United States and in a variety of geographic and social settings. In this paper, we discuss the potential as well as the limitations we found in application to the NPS visitor use statistics as well as more generally for application to unmonitored public lands.

## Materials and methods

### Visitation data

The NPS Social Science Program coordinates visitation statistics for almost 400 park units. Though the time window of visitation data varies by park, the data are available at the monthly resolution from 1979 to the present calendar year for most parks. Counting visitors often requires a high degree of expertise in social and statistical science as well as in-depth knowledge of a particular park to tailor an approach for data collection and for adjusting raw counts

[13]. To ensure the consistency and reliability of data, the program staff work with individual parks to develop procedures for data collection that best fit to each park unit. This effort also includes detailed documentation on any potential factors that could have impacts on monthly visitation counts. All the information is publicly accessible through the NPS Stats database (https://irma.nps.gov/STATS/).

We obtained the park boundaries from the DataStore at the NPS website (https://irma.nps.gov/DataStore/) and downloaded monthly estimates of recreational visits, excluding workers or pass-through residents, for the top 50 most visited parks in years 2018 and 2019, excluding National Parkways (e.g., Blue Ridge Parkway). We excluded parkways because our data provider identified a visit based on the estimated dwell time of a device of more than five minutes. For parkways it is likely that non-park visitors (e.g., pass-through drivers) would also be identified as park visitors when sitting in traffic. We inspected the validity of the monthly records using the comments for each month provided by the NPS park which flagged potential problematic records due to data collection issues (e.g., impaired counters, power outages, staffing limitations, maintenance work, and government shutdowns). We excluded any park for which we were left with fewer than 10 monthly values after removing flagged records likely to impact monthly park visitation counts. We also removed Stonewall National Monument because its high visitation was driven by a single event (i.e., World Pride event in June 2019) and was closed for half of the time in years 2018 and 2019. We also removed Golden Gate National Recreation Area after realizing the cell data were likely capturing non-visitor drivers and the NPS data were not, given the road network contained within the park boundary. After removing these flagged values and parks, we were left with a total of 38 parks (Fig 1) with 786 monthly visitation records. Total visitation at these 38 parks accounts for around 43.1% and 42.6% visitation at all the NPS units in 2018 and 2019, respectively.

Using the official park boundaries [47], we purchased the cell data from Airsage Inc. representing visitation to those areas, specifically using their Target Location Analytics (TLA) product. This provider creates population-level estimates derived from around a third of the U.S. population [40 for more details]. For each park unit, the data consist of hourly, daily, and monthly estimates of visitation based on Airsage's algorithm that expands their sample of devices to the total population. We do not have access to the device-level information, but rather aggregated by time-window and park unit (point of interest [POI] in Airsage's terminology). In addition to visitation by time, the dataset includes visitor origin information consisting of the count of unique visitors by month to each POI by Census block group (CBG).

The location data come from smartphone applications and their users who choose to opt-in to location sharing on location-based services. These device-level locations, derived from GPS data, are processed by Airsage to create visits based on the behavior of the device. According to the provider, the datasets represent information derived from the locations of over 120 million devices each month, or about a third of the U.S. population.

The home and work location of each device is defined by its behavior during the month. Home is defined to be the census block group where each device is most frequently located between the hours of 9PM and 6AM. Work is defined as where each device is most frequently located between the hours of 8AM and 4PM. Any other device location is defined as an activity point. The data we used are for activity points (non-work, non-home) within the geographies of the selected parks.

The Airsage TLA data came with two general file formats. One of these contains the hourly and daily visitation to each POI for each month. These are referred to as the POI summary files. We used these files to examine the relationship to NPS counts by summing the daily visits to each park for each month form the POI summary file. For our purposes, each POI corresponds to one NPS unit. The second, at a monthly resolution, the "HOME" file summarizes

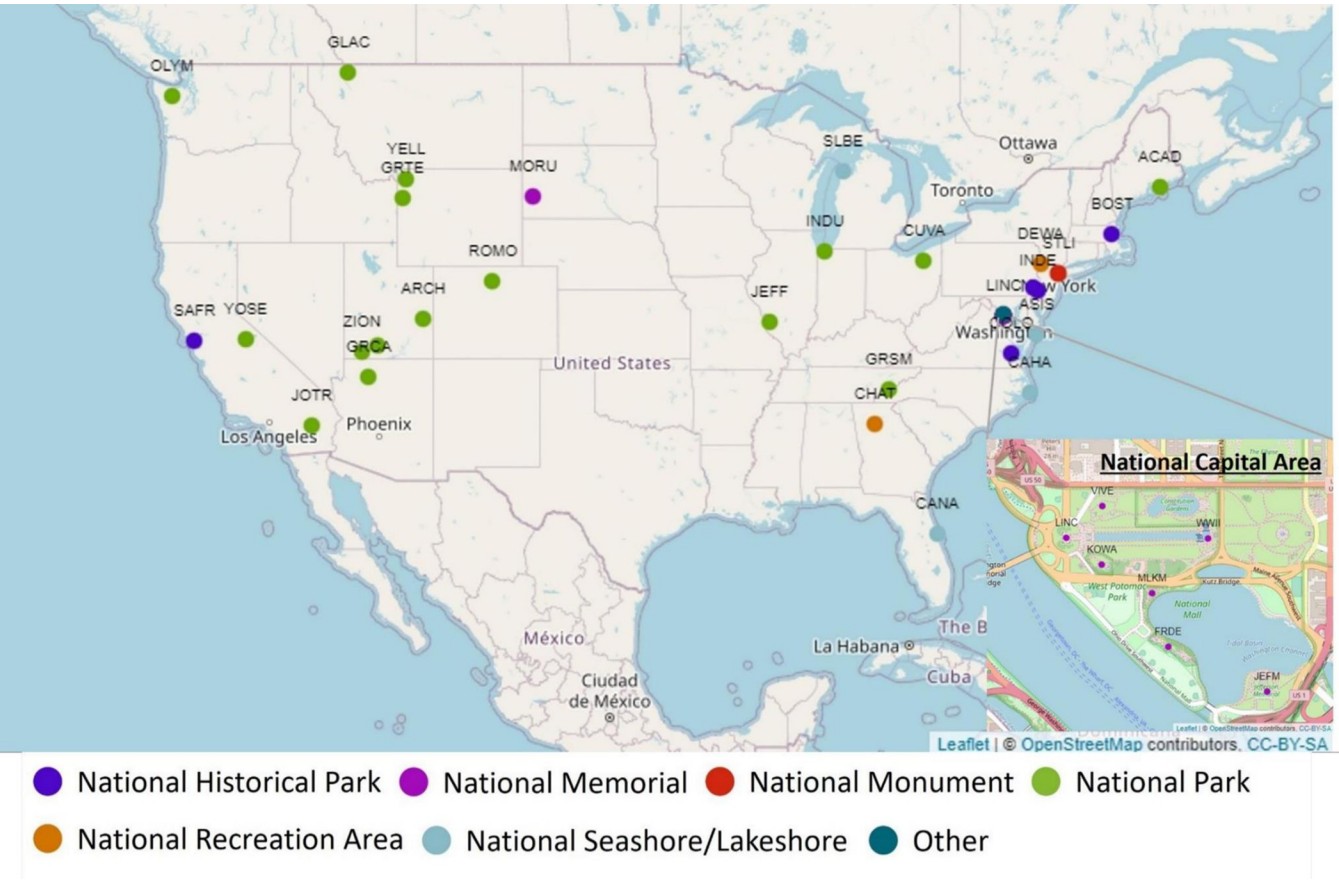

**Fig 1. Distribution of the selected park units.** The four-letter code for each park can be found in Table 1. The coordinates of NPS locations were obtained from the NPS Data Store (https://irma.nps.gov/DataStore/) for public access with a proprietary/copyright designation of "No Restriction - The information is neither copyrighted nor has any other use restrictions related to it being intellectual property. For that reason, this information may be distributed to the NPS and public" and plotted using leaflet package in R 4.2.2. Base map tiles are from OpenStreetMap and OpenStreetMap Foundation. This map contains information from OpenStreetMap and OpenStreetMap Foundation, which is made available under the Open Database License, CC BY-SA (https://www.openstreetmap.org/copyright).

the count of visitors by home CBG of origin to each POI. We used this file to calculate the travel distance between each home CBG and each park for further classification in this analysis (described in Park settings section).

A summary of the data sources and statistics of total visitation used in this analysis are described in S1 Table.

## Park attributes

Since parks use various counting methods and their geographic and physical settings differ widely, we hypothesized three park attributes that might explain differences in the relationship between the cell data and the NPS recreation visit estimates. These attributes included park types, park settings, and porousness.

**Park type.** The NPS park type is an official designation. The selected 38 parks included the following types: national park (N = 16), national recreation area (N = 2), national memorial (N = 8), national monument (N = 2), national seashore (N = 3), national historical park (N = 5), national lakeshore (N = 1), and 'other' (N = 1). Since only one unit is designated as a national lakeshore, we combined it with national seashore units. Rock Creek Park was the only park in the 'Other' category.

**Park setting.** We hypothesized that the interactions between park distance from the population center (i.e., urban or non-urban) and recognition from non-local communities (i.e., iconic or local) of a park might play a role in the relationship between cell data counts and NPS counts. Urban and non-urban settings may provide different opportunities for recreational activities (e.g., accessibility, type of recreation, carrying capacity) have different visitation profiles, and use different counting methods. We identified urban versus non-urban parks using the NPS definition of population centers for "Urban Area Park" [48], which is defined as having more than 75% of total area located within an "Urbanized Area" defined by the 2010 Census Urban and Rural Classification and Urban Area Criteria (Census Urban Area hereafter). We calculated the proportion of park area within the Census Urban Area by overlaying the polygons of selected parks and the Census Urban Area.

In addition to the differences by proximity to urban area, we theorized that parks that are well-known beyond the local area, such as Yosemite National Park and the Lincoln National Memorial, may employ different counting methods and may even receive more resources devoted to NPS efforts to count visitation compared to parks that are mostly visited by local communities, such as Rock Creek Park. We classified parks into iconic or local type respectively for those recognized by non-local visitors and those mostly known by local communities using the proportion of the local visitors. We calculated Euclidean distance from the centroid of the POI (i.e., parks) to home CBGs and defined any pair of distance within 50 miles as local commuting areas based on the definition of local visitor used in the U.S. Forest Service National Visitor Use Monitoring program [49]. Parks with a proportion greater than 30% of the visitors coming from the local commuting areas were defined as "local" type (Fig 2). However, this criterion likely caused a false 'local' classification for parks close to highly populated cities, where a high percentage of local residents may visit nearby parks often. We revised these parks (i.e., Olympic National Park, Statue of Liberty National Monument, Thomas Jefferson Memorial, and World War II Memorial) to iconic parks.

We created four groups based on the interactions between Urban/Non-Urban and Iconic/Local to represent four types of parks with the main characteristics being:

- Urban & Iconic: has international name-recognition and/or is frequently visited as part of a trip to an urban destination; Many of the Washington D.C. memorials fit into this category.

- Urban & Local: has less name-recognition and is visited primarily by local residents in an urban setting; Rock Creek Park in Washington D.C. fits in this category.

- Non-Urban & Iconic: has international name-recognition and/or is usually a main destination for a vacation that is far from populated areas; Yosemite and Yellowstone National Park fit in this category.

- Non-Urban & Local: has less name-recognition and is visited primarily by local communities in a remote setting; Many of the historic parks and seashores fit into this category.

**Porousness.** Porousness refers to the potential park entry points, including official and unofficial publicly accessible entrances. We theorized that porousness would impact the method used for the on-the-ground data collection and accuracy of the NPS visitor counts, because adding entry points makes it more difficult to count all visitors. Entry points to a park are usually via automobile roads and foot traffic trails, although some parks (e.g., the Lincoln Memorial) have almost continuously porous boundaries. We considered the intersections of park boundaries with roads and trails as potential entry points and obtained these intersections by overlapping the road and trail segments that are identified with public access in the NavTEQ Streets database [50] with the official park boundaries. We further removed points within

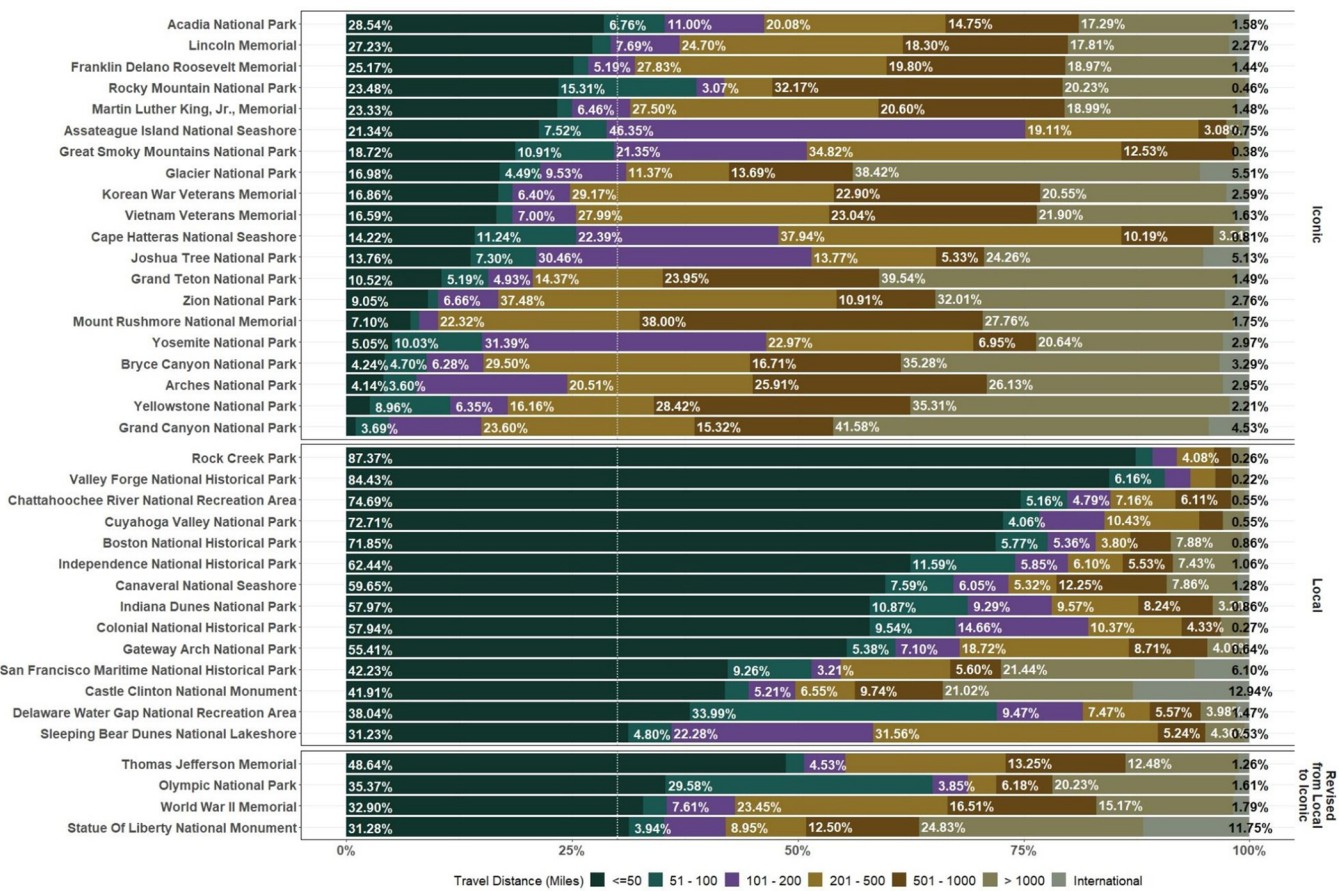

**Fig 2. Categories of travel distance between origins and destinations for identifying iconic or local parks.** A park with greater than 30% of the visitors travelled within 50 miles is identified as local type, otherwise iconic type. Some exceptions are applied (Revised from Local to Iconic).

10m of an adjacent point since they likely indicate the same entrance. We considered that parks with less than or equal to three total intersections as low porousness. The number of intersections of roads and trails with park boundaries ranged from 0 for the Statue of Liberty National Monument, which is on an island, to 113 in the Delaware Water Gap National Recreation Area. Five parks (Korean War Veterans Memorial, Lincoln Memorial, Mount Rushmore National Memorial, Vietnam Veterans National Memorial, and Statue of Liberty National Monument) have less than or equal to 3 intersections.

For parks with greater than three intersections, we used intersection density (i.e., total number of intersections divided by total area of the park) to classify parks into low and high porousness. We considered that porousness for small and large park units may function differently. Most of the small park units are in populated cities, where more trails, roads, and sidewalks can intersect the park unit, and which usually have small viewshed areas. In contrast, the intersections of roads and trails with boundaries in big parks are often difficult to capture from one observation point. Therefore, we defined low and high levels of porousness for small (total area $\leq$ 1 square kilometer) and large parks using the respective median values of the intersection density as the thresholds.

Thirteen parks are smaller than 1 square kilometer, with a median value of intersection density is 100 entry points per square kilometer. Among these 13 parks, 8 units are classified as low porousness and 5 are classified as high porousness. For large parks (25 units), a total of 14

and 11 units each are classified as low and high levels of porousness, respectively, after taking the median value of intersection density (0.04 entry points per square kilometer) into account.

A total of four porousness levels, Small-Low, Small-High, Large-Low, and Large-High, were used as park attributes for analysis.

## Statistical approach

To understand the associations between the NPS' visitation data and the cell data, we tested bivariate and multivariate relationships. We applied Pearson's correlations to examine the bivariate relationships between NPS counts and cell data counts. We also attempted to observe whether there were similarities in correlations by groupings of parks described in Park attributes (e.g., park type, park setting, and porousness).

We used a panel regression model to examine multivariate relationships between NPS counts and cell data counts. We then stratified the analysis by park setting and porousness but not park type, due to the available data points in each grouping. We applied a mixed-effects model to account for potential clustering by NPS unit, and specified it to incorporate random effects across parks, to produce a model that may be appropriate to generalize to a park outside of this sample of parks, as opposed to using park-level fixed effects. Using a random effect specification, we estimated the following model:

$$log\ (Y_{it}) = \beta_c log\ (Cell_{it}) + \beta_m M + \mu_i + e_{it}$$

where,

$Y_{it}$ - Monthly visitation count from NPS data for park $i$ and month $t$

$Cell_{it}$ - Cell data count for park $i$ and month $t$

$M$ - Vector of month

$\mu_i$ - Between park error term – the random effect

$e_{it}$ - Within park error term

We included month as a variable to investigate how these relationships vary by month. We hypothesized that the NPS count methods within parks may vary by month due to staffing and different seasonal use and that the cell data may perform differently based on months due to the sample size and/or types of users.

We specified a log-log relationship between the cell data and the NPS counts to account for the differences in the scales of visitation among parks. The coefficients on the cell data counts can be interpreted as a given percent change in the cell data count (the regressor) resulting in a $\beta$% change in the NPS counts (response variable). In the case of the month variables, the NPS count changes by $100*exp(\beta_m-1)$% for a given month. While marginal R-squared values consider fixed effects only, the multivariate relationships were primarily evaluated by conditional R-squared values, which take both fixed and random effects into account.

## Results

Comparing mean monthly visitation for 2018 and 2019 by park type (Fig 3), we found NPS visitation counts to be generally lower than the raw cell data counts, except for National Memorials and National Monuments, where the NPS counts were higher. National Seashores/Lakeshores, National Monuments, and National Parks show peaks of visitation in summer (June, July, and August) in both NPS and cell data, whereas National Memorials have higher visitation in spring (March, April, and May). The greatest differences between the NPS and cell visitation counts occurred for National Recreation Areas, with a mean difference of -2,598,096 (standard deviation [sd] = 977,138) visits when using NPS data as the base. Visitor

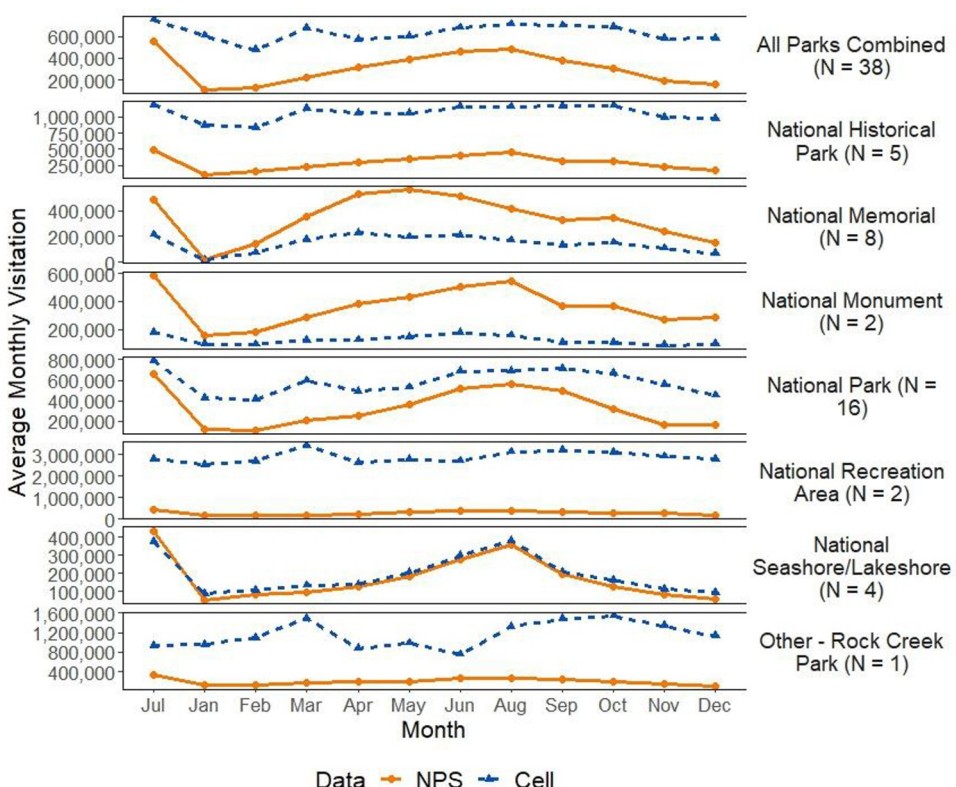

**Fig 3. Average monthly visitation using NPS counts and cell data by park type for years 2018 and 2019.**

counts for National Seashores/Lakeshores were most closely aligned between NPS and cell visitation counts (mean difference = -21,060, sd = 96,372). Cell visitation counts and NPS counts appeared to be closely aligned for some individual parks (e.g., Yellowstone, Arches, Acadia, Grand Teton; S1 Fig), aligned in the winter months but biased during summer for other parks (e.g., Rocky Mountain, Glacier), and biased throughout the year for national recreation areas and national historic parks.

## Bivariate relationships

Bivariate relationships showed that 20 parks had a high correlation ($> 0.8$) and 8 parks had a low correlation ($< 0.5$) between monthly Cell data and NPS counts (Table 1). Fig 4 presents observed correlations by various groupings, summarizing the range of correlations. Correlations by park type showed that National Memorials and National Parks tended to have better correlations between NPS counts and cell data, with a respective mean correlation of 0.86 (sd = 0.07) and 0.82 (sd = 0.20). However, correlations were low for National Recreation Areas (0.26, sd = 0.10) and Other – Rock Creek Park (-0.16).

When grouping by urban location and iconic status, higher correlations were observed for iconic parks than for local parks, with mean correlations of 0.84 (sd = 0.06), 0.91 (sd = 0.09), 0.39 (sd = 0.33), and 0.53 (sd = 0.21) for Urban & Iconic, Non-Urban & Iconic, Urban & Local, and Non-Urban & Local, respectively.

Grouped by park size and porousness, we found that large parks with low porosity generally had higher correlations, with a mean correlation of 0.88 (sd = 0.17), whereas large parks with high porosity had lower correlations (mean = 0.53, sd = 0.30). NPS counts and cell data were

**Table 1. Park attributes.**

| NPS Code | Park Name | Park Type | Number of Records | Population Center and Recognition Level | Porousness Level | Pearson's Correlation |
|---|---|---|---|---|---|---|
| BOST | Boston National Historical Park | National Historical Park | 23 | Urban & Local | Small Park - Low Porosity | 0.730 |
| COLO | Colonial National Historical Park | National Historical Park | 23 | Non-Urban & Local | Large Park - High Porosity | 0.542 |
| INDE | Independence National Historical Park | National Historical Park | 24 | Urban & Local | Small Park - High Porosity | 0.743 |
| SAFR | San Francisco Maritime National Historical Park | National Historical Park | 24 | Non-Urban & Local | Small Park - Low Porosity | 0.496 |
| VAFO | Valley Forge National Historical Park | National Historical Park | 17 | Urban & Local | Large Park - High Porosity | 0.221 |
| FRDE | Franklin Delano Roosevelt Memorial | National Memorial | 20 | Urban & Iconic | Small Park - High Porosity | 0.928 |
| KOWA | Korean War Veterans Memorial | National Memorial | 20 | Urban & Iconic | Small Park - Low Porosity | 0.852 |
| LINC | Lincoln Memorial | National Memorial | 21 | Urban & Iconic | Small Park - Low Porosity | 0.792 |
| MLKM | Martin Luther King, Jr., Memorial | National Memorial | 20 | Urban & Iconic | Small Park - High Porosity | 0.859 |
| MORU | Mount Rushmore National Memorial | National Memorial | 24 | Non-Urban & Iconic | Large Park - Low Porosity | 0.988 |
| JEFM | Thomas Jefferson Memorial | National Memorial | 20 | Urban & Iconic | Small Park - Low Porosity | 0.897 |
| VIVE | Vietnam Veterans Memorial | National Memorial | 20 | Urban & Iconic | Small Park - Low Porosity | 0.785 |
| WWII | World War II Memorial | National Memorial | 20 | Urban & Iconic | Small Park - High Porosity | 0.817 |
| CACL | Castle Clinton National Monument | National Monument | 24 | Urban & Local | Small Park - High Porosity | 0.621 |
| STLI | Statue Of Liberty National Monument | National Monument | 24 | Urban & Iconic | Small Park - Low Porosity | 0.750 |
| ACAD | Acadia National Park | National Park | 13 | Non-Urban & Iconic | Large Park - High Porosity | 0.975 |
| ARCH | Arches National Park | National Park | 22 | Non-Urban & Iconic | Large Park - Low Porosity | 0.876 |
| BRCA | Bryce Canyon National Park | National Park | 21 | Non-Urban & Iconic | Large Park - High Porosity | 0.917 |
| CUVA | Cuyahoga Valley National Park | National Park | 22 | Non-Urban & Local | Large Park - High Porosity | 0.527 |
| JEFF | Gateway Arch National Park | National Park | 21 | Urban & Local | Small Park - Low Porosity | 0.387 |
| GLAC | Glacier National Park | National Park | 23 | Non-Urban & Iconic | Large Park - Low Porosity | 0.996 |
| GRCA | Grand Canyon National Park | National Park | 16 | Non-Urban & Iconic | Large Park - Low Porosity | 0.752 |
| GRSM | Grand Teton National Park | National Park | 13 | Non-Urban & Iconic | Large Park - Low Porosity | 0.973 |
| GRTE | Great Smoky Mountains National Park | National Park | 24 | Non-Urban & Iconic | Large Park - High Porosity | 0.679 |
| INDU | Indiana Dunes National Park | National Park | 23 | Non-Urban & Local | Large Park - High Porosity | 0.468 |
| JOTR | Joshua Tree National Park | National Park | 10 | Non-Urban & Iconic | Large Park - Low Porosity | 0.867 |

*(Continued)*

**Table 1.** (Continued)

| NPS Code | Park Name | Park Type | Number of Records | Population Center and Recognition Level | Porousness Level | Pearson's Correlation |
|---|---|---|---|---|---|---|
| OLYM | Olympic National Park | National Park | 22 | Non-Urban & Iconic | Large Park - Low Porosity | 0.902 |
| ROMO | Rocky Mountain National Park | National Park | 17 | Non-Urban & Iconic | Large Park - Low Porosity | 0.989 |
| YELL | Yellowstone National Park | National Park | 22 | Non-Urban & Iconic | Large Park - Low Porosity | 0.986 |
| YOSE | Yosemite National Park | National Park | 23 | Non-Urban & Iconic | Large Park - Low Porosity | 0.885 |
| ZION | Zion National Park | National Park | 20 | Non-Urban & Iconic | Large Park - Low Porosity | 0.946 |
| CHAT | Chattahoochee River National Recreation Area | National Recreation Area | 24 | Urban & Local | Large Park - High Porosity | 0.192 |
| DEWA | Delaware Water Gap National Recreation Area | National Recreation Area | 23 | Non-Urban & Local | Large Park - High Porosity | 0.335 |
| ASIS | Assateague Island National Seashore | National Seashore/Lakeshore | 24 | Non-Urban & Iconic | Large Park - Low Porosity | 0.898 |
| CAHA | Canaveral National Seashore | National Seashore/Lakeshore | 23 | Non-Urban & Local | Large Park - Low Porosity | 0.367 |
| CANA | Cape Hatteras National Seashore | National Seashore/Lakeshore | 20 | Non-Urban & Iconic | Large Park - High Porosity | 0.853 |
| SLBE | Sleeping Bear Dunes National Lakeshore | National Seashore/Lakeshore | 24 | Non-Urban & Local | Large Park - High Porosity | 0.966 |
| ROCR | Rock Creek Park | Park (Other) | 12 | Urban & Local | Large Park - High Porosity | -0.157 |

generally correlated well for small parks, with mean correlations of 0.71 (sd = 0.17) and 0.79 (sd = 0.11) for low and high porousness, respectively.

## Regression

Across all parks, a 1% increase in the cell data counts corresponded to a 0.86% increase in the NPS counts (Table 2) and the model fitted well in terms of the conditional R-squared value (0.96). However, this relationship between cell data and NPS counts varied from 0.3% to 1.0% across our stratifications, with only the urban and local grouping not having a statistically significant relationship. Looking across stratifications, the non-urban and iconic group and the large and low porousness had cell data coefficients close to 1. All the stratified models fitted well, with conditional R-squared values in the 90% range. The model performance evaluated by the plot of NPS counts versus predicted values (transformed back to levels from logs) showed that the distribution was close to the one-to-one line (Fig 5).

There seemed to be a consistent pattern in the relationship between the cell data and NPS counts by month (S2–S4 Tables). The monthly coefficients implied that an increase in cell data counts would correspond to a smaller increase in NPS counts for winter months compared to the month of July (reference) and other months in summer. In other words, it took more cell data counts to correspond to a similar NPS count in the winter than in the summer in consistent seasonal pattern.

## Discussion

Collecting visitation information across large spatial and time scales in a consistent manner is challenging [14, 51]. While cellular location data have been applied to track human mobility

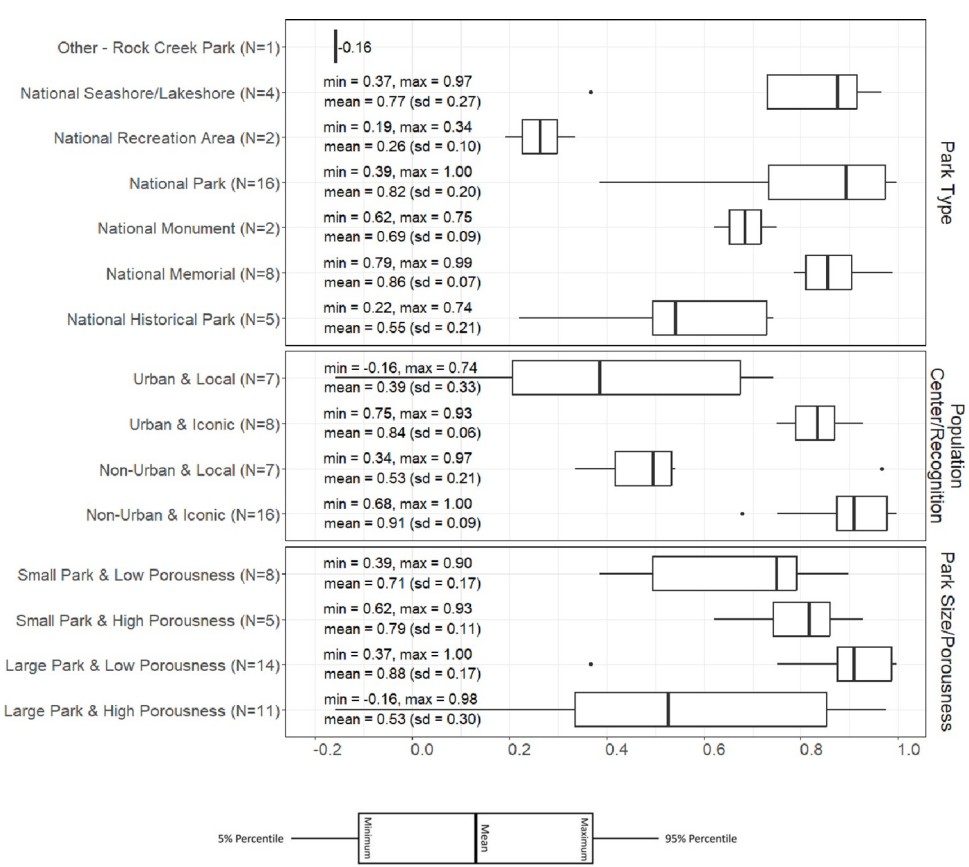

**Fig 4. Correlations between cell and NPS data grouped by park type, population center/recognition, and park size and porousness.** Lines extends to 5th and 95th percentiles.

patterns during the COVID-19 pandemic [36, 37, 52], the use of cell phone data to understand how many people visit recreational resources is increasing but mostly has been limited to a single geographic region and setting [38–41, 53, 54]. This study shows that the cell data product can predict NPS counts gathered from broad geographic areas well, as shown by bivariate correlations and regression modeling. This supports the feasibility of using these products to

**Table 2. Regression results.** CIs stands for Confidence Intervals.

| Model | Coefficient of Cell data | 95% CIs | p | Number of NPS Units | Number of Records | Marginal/ Conditional R-squared value |
|---|---|---|---|---|---|---|
| All Parks | 0.863 | 0.809, 0.917 | <0.001 | 38 | 786 | 0.555 / 0.964 |
| **Stratification by Population Center and Recognition Level** | | | | | | |
| Urban & Iconic | 0.330 | 0.255, 0.405 | <0.001 | 8 | 165 | 0.721 / 0.954 |
| Urban & Local | 0.058 | -0.129, 0.245 | 0.561 | 7 | 145 | 0.507 / 0.820 |
| Non-Urban & Iconic | 1.011 | 0.940, 1.081 | <0.001 | 16 | 314 | 0.880 / 0.959 |
| Non-Urban & Local | 0.913 | 0.725, 1.101 | <0.001 | 7 | 162 | 0.659 / 0.934 |
| **Stratification by Park Size and Porousness** | | | | | | |
| Large & Low | 1.024 | 0.943, 1.106 | <0.001 | 14 | 280 | 0.853 / 0.937 |
| Large & High | 0.883 | 0.748, 1.018 | <0.001 | 11 | 225 | 0.600 / 0.949 |
| Small & Low | 0.285 | 0.142, 0.427 | <0.001 | 8 | 173 | 0.396 / 0.911 |
| Small & High | 0.402 | 0.305, 0.500 | <0.001 | 5 | 108 | 0.743 / 0.954 |

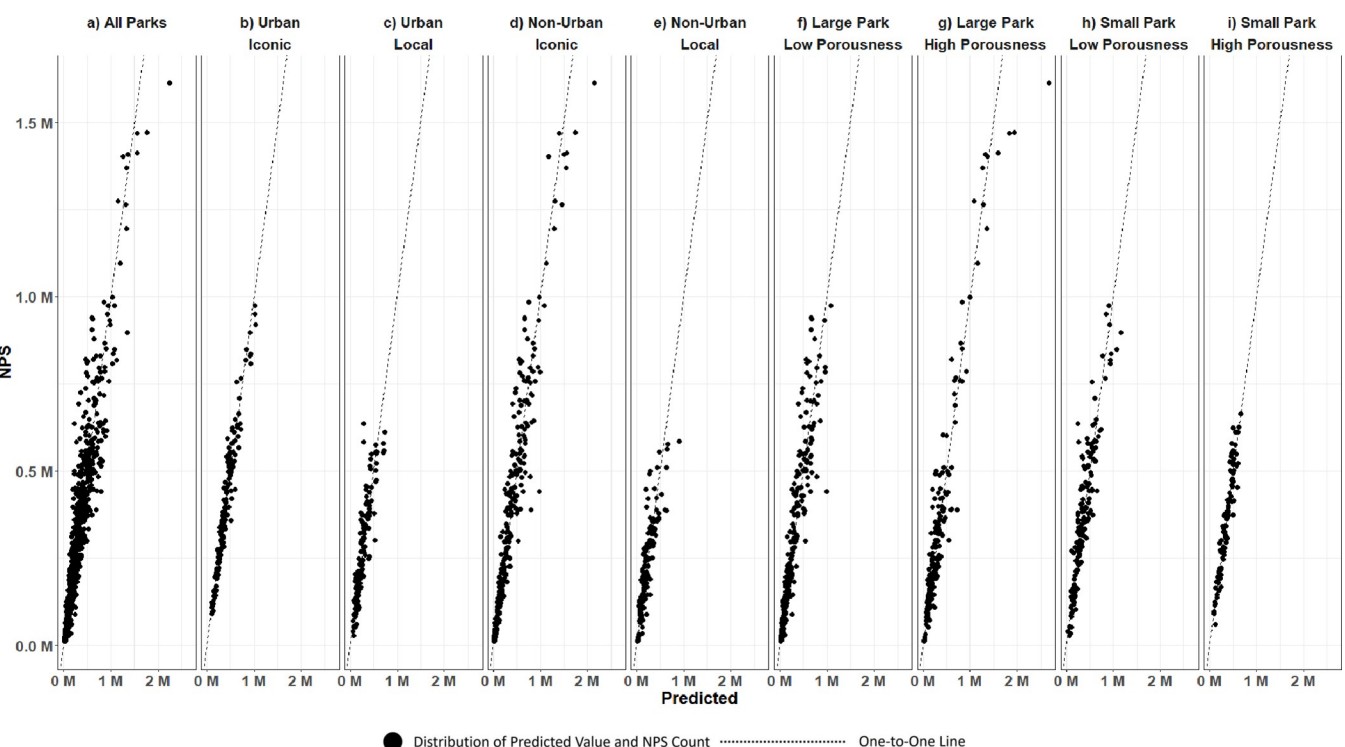

**Fig 5.** Model performance evaluated by NPS counts versus predicted values from the models for all parks combined (a), and stratification by population center and recognition from non-local community (b to e) and by size and porousness level (f to i).

complement on-the-ground methods for estimating park visitation, similar to recent findings by others [38, 39, 53, 54]. However, we found the strength and magnitude of the relationship between NPS counts and cell data varied by the groupings we used and time of year, which poses questions related to using cell data products but also highlights potential issues and improvement needs of the current NPS visitor quantification system. Below we discuss limitations of using cell data for park visitation analysis and limitations of this study, as well as future research directions.

The relationship between on-the-ground counts and cell data is strong for many parks and weak for others. This relationship could vary for many reasons including unknown issues with the cell phone data. However, our analysis may imply that the accuracy of NPS counts varies depending on park attributes (e.g., park type, porousness, urbanicity, level of recognition, counting method, etc.) or could even vary across individual parks within these categories. For instance, corresponding to previous studies [14, 38, 39], we found proximity of population center, recognition from non-local communities, and porousness of a park affected the correlations between on-the-ground counts and cell data. We investigated the potential variations in the relationships between NPS and cell data counts by park attributes we felt were important and for which we could compile data. We were not able to find data for all attributes we would have liked to examine. For example, we would have liked to include park access transportation mode (e.g., walking, private vehicle, bus, biking, horseback, ferry, etc.), but the NPS does not collect these data. Park access mode is somewhat reflected by the NPS counting methods. However, we found that most of the parks use a combination of counting methods and do not provide visitor counts by each counting method. On-the-ground counting data collections are usually complex (e.g., a combination of traffic counter with various adjustments for person-

per-vehicle at different entrances and/or seasons, cars in the camping sites, and assuming a constant number per bus).

The variations in the relationships between NPS and cell data counts may also be due to physical factors leading to inaccurate extrapolation from the available source of visitation records that are being used by NPS (e.g., Person-Per-Vehicle, which assumes number of people per car) that have not been revisited for several years, or differences in how the NPS count methods are applied based on variations in the type of use by month. If we assume the visitation counts collected by cell data are consistent across parks, cell data may be used to investigate visitor quantification methods and harmonize the visitor quantification methods across parks or months. For example, we found consistent differences by month in the relationship between cell data and NPS counts which may indicate variations in the accuracy of NPS visitation counting methods by season (see supplementary material, S2–S4 Tables). However, the differences in the relationships could also be due to variation in how the cell data performs in the winter months when there may be a lower number of visitors.

Although it is promising that, for many parks (i.e., 20 parks had a correlation > 0.8), the relationship between the NPS and cell data counts is strong in our analysis, we cannot disentangle whether the NPS or the cell data counts are closer to the true visitation value for any given park in this study. Both sources of visitation information have potential biases and inaccuracies. Though we used the official park boundaries defined by the NPS, it is possible that there are discrepancies between the real spatial extent used by the NPS for counts versus those we used as POI to obtain visitation using the cell phone devices. While we understand many of the limitations of on-the-ground count methods, less is understood about the limitations of the cell phone data. One potential source of variation, for example, is that the NPS counts are designed to measure park visitation versus people who are using the roads merely to get from point A to point B or people who are employed or delivering goods within the park. Although the cell phone data only included devices that spent more than five minutes at any location within the park boundary, we likely captured a significant number of employees and people using park roads for transportation rather than for visiting a park. The representativeness of the cell phone device sample, or the likelihood visitors having their devices turned on and an application providing location data while in the park could also contribute to accuracy issues. Further research using ground-truth data is needed to disentangle these mechanisms.

The commercial cell data location industry is varied as to their products, customers and intended uses. Their data compilation and processing techniques are generally not transparent to their buyers and can change over time, even during a study period. The specific apps used for the locational information and the algorithms applied have been treated as protected business information. Ground-truthing is often not built into data provider processes, or the ground-truth and model fitting was conducted with similar metrics, like vehicle traffic, but not a direct comparison to metrics of interest like foot traffic, or more specific metrics like unique daily visits. Data users need to be aware of these aspects that can impact data quality. For example, the stability of device location samples used by a cell data provider, sourced from a collection of smartphone applications, could change over the period of a study leading to changes in algorithms and potential changes in data quality. Our approach has been to verify against contemporaneous on-the-ground visitation counts for the timeframes of interest in our studies.

The practical result of this conundrum is that cell data, at this point, may be a useful tool for enhancing existing visitation quantification programs but cannot replace the on-the-ground methods entirely. It could, for example, be used to fill in or impute visitation estimates for missing months or for parks where counts were halted due to broken equipment or staffing issues. We found and removed from the dataset many of the months flagged by

the NPS as being inaccurate for various reasons. At the expense of some accuracy, cell data may also allow for a scaled back on-the-ground sampling scheme in parks where cell data performs well and where the stability of that relationship could be repeatedly tested over time. It may also be useful to capture use in areas of a park where on-the-ground counts are impractical, such as on remote Bureau of Land Management lands where counts are not conducted, or on city parks with many entrances and lack of resources for surveys. In addition, information related to visitors' experiences still require detailed surveys, which cell data alone will not be able to provide.

For natural lands and parks without any estimates of visitation–the vast majority–cell data may provide an acceptable estimate of visitation when used in conjunction with a model built from relationships to observed visitation series. Once the cell data are calibrated against data from parks that do have an existing on-the-ground sampling program, such as the NPS, estimates of the level of visitation with ranges of uncertainty could be created for unmonitored sites. Creating and testing a more general and transferrable function using cell data to predict visitation for monitored and unmonitored park lands is left for future work, but the descriptive model results using the NPS visitation program provided by this paper are promising.

## Conclusions

Understanding human use of parks and other protected areas is important for managers to evaluate social, economic, ecological, and environmental values. Crowd-sourced data offer an alternative approach to the challenge of collecting visitation counts consistently across large scales of geography and time via traditional ground counting methods. While cell data are an exciting addition and a potential scale-multiplier (i.e., 1 cell data count equals to x on-the-ground count) for on-the-ground visitation quantification methods, it is not a replacement. Finding opportunities to calibrate and validate these new sources of data on human behavior and movement is still needed for practitioners to have confidence and defendable methods to supplement their current visitor quantification programs. For places without estimates of visitation, a cell data model calibrated to a known visitor use monitoring program may provide information on the use of natural lands using a scalable and repeatable instrument.

## Supporting information

**S1 Table. Data sources and summary statistics of total visitation and visitation by month for years 2018 & 2019 for 38 parks used in the analysis.**
(DOCX)

**S2 Table. Regression model for all parks combined.** CIs stands for Confidence Intervals.
(DOCX)

**S3 Table. Stratified analysis by population center and iconic status.** CIs stands for Confidence Intervals.
(DOCX)

**S4 Table. Stratified analysis by park size and porousness level.** CIs stands for Confidence Intervals.
(DOCX)

**S1 Fig. Average monthly visitation using NPS counts and cell data for each park for years 2018 and 2019.**
(DOCX)

## Acknowledgments

The authors would like to thank Dr. Kate Mulvaney, Dr. Marisa Mazzotta, and Erin Burman (EPA) Dr. Spencer Wood and Sama Winder (University of Washington), Dr. Eric White (US Forest Service), and Dr. Emily Wilkins (US Geological Survey) for their time and thoughtfulness in providing valuable feedback. We would also like to express our gratitude to Dr. Pamela Ziesler for providing insights on the NPS visitation count data and to Dr. Maliha Nash for her statistical advice. This paper has been reviewed by the Center for Public Health and Environmental Assessment in the Office of Research and Development at the U.S. Environmental Protection Agency and approved for publication. Approval does not signify that the contents reflect the views and policies of the Agency, nor does the mention of trade names of commercial products constitute endorsement or recommendation for use.

## Author Contributions

**Conceptualization:** Wei-Lun Tsai, Nathaniel H. Merrill, Anne C. Neale, Madeline Grupper.

**Data curation:** Wei-Lun Tsai, Nathaniel H. Merrill, Anne C. Neale, Madeline Grupper.

**Formal analysis:** Wei-Lun Tsai, Nathaniel H. Merrill, Anne C. Neale.

**Funding acquisition:** Nathaniel H. Merrill, Anne C. Neale.

**Investigation:** Nathaniel H. Merrill, Anne C. Neale.

**Methodology:** Wei-Lun Tsai, Nathaniel H. Merrill, Anne C. Neale.

**Project administration:** Wei-Lun Tsai, Nathaniel H. Merrill.

**Resources:** Nathaniel H. Merrill, Anne C. Neale.

**Supervision:** Nathaniel H. Merrill, Anne C. Neale.

**Validation:** Wei-Lun Tsai, Nathaniel H. Merrill, Anne C. Neale.

**Visualization:** Wei-Lun Tsai, Nathaniel H. Merrill.

**Writing – original draft:** Wei-Lun Tsai, Nathaniel H. Merrill, Madeline Grupper.

**Writing – review & editing:** Wei-Lun Tsai, Nathaniel H. Merrill, Anne C. Neale, Madeline Grupper.

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
