## [Decision Letter · Decision Letter 0]

13 Jun 2023

PONE-D-23-15317Using cellular device location data to estimate visitation to public lands: An example comparing device location data to U.S. National Park Service’s visitor use statisticsPLOS ONE

Dear Dr. Tsai,

Thank you for submitting your manuscript to PLOS ONE. Both of our reviewers have raised a few ways to improve the paper.  Therefore, we invite you to submit a revised version of the manuscript that addresses the points raised during the review process.

Please note that PLOS uses the reference style outlined by the International Committee of Medical Journal Editors (ICMJE), also referred to as the “Vancouver” style. Example formats are listed below. Additional examples are in the ICMJE sample references. Please make sure the reference list confines with the journal's requirement. 

You can see each reviewer's feedback, please address these comments in your revised paper. 

We look forward to receiving your revised manuscript.

Kind regards,

Zihao Zhang

Academic Editor

PLOS ONE

Journal Requirements:

Additional Editor Comments:

PLOS uses the reference style outlined by the International Committee of Medical Journal Editors (ICMJE), also referred to as the “Vancouver” style. Example formats are listed below. Additional examples are in the ICMJE sample references.

Reviewers' comments:

Reviewer's Responses to Questions

**Comments to the Author**

1. Is the manuscript technically sound, and do the data support the conclusions?

Reviewer #1: Yes

Reviewer #2: Yes

2. Has the statistical analysis been performed appropriately and rigorously? 

Reviewer #1: Yes

Reviewer #2: Yes

3. Have the authors made all data underlying the findings in their manuscript fully available?

Reviewer #1: No

Reviewer #2: No

4. Is the manuscript presented in an intelligible fashion and written in standard English?

Reviewer #1: Yes

Reviewer #2: Yes

5. Review Comments to the Author

Reviewer #1: Thank you for the opportunity to review your manuscript. The study explored the use of commercially available cellular data for estimating visitation to public lands by comparing it with NPS visitor use statistics. It is a well-written, needed, and useful study of the use of cell data. The manuscript effectively summarizes the methods and results. Here are some recommendations:

• In the introduction section, the manuscript needs to emphasize the unique contribution of this study. On page 3, the authors listed many existing studies that found cell data and on-site counts were comparable. What are the research gaps? Why is this study needed? What are the theoretical, methodological, or empirical contributions of this study?

• The study explored the role of park type, park setting, and porousness in explaining differences between the cell data and NPS recreation visit estimates. There might be some other important factors. For example, geographic characteristics might influence the quality of cell data. The proportion of different modes to access park units, such as private vehicles, biking, walking, bus, and boat, might affect the NPS counts. If these data are not available, these issues need to be discussed as limitations.

• On page 10, the manuscript states, “the cell data may be appropriate for trends analysis.” Although this thought is interesting and valuable, more data and evidence are needed to support this argument.

• Page 10, the manuscript states, “It may also be useful to capture use in areas of a park where on-the-ground counts are impractical.” More explanations are needed to describe the situations when on-the-ground counts are impractical.

• Figure 2, in each category, parks are ordered alphabetically. Reordering the parks based on the proportion of local visitors (i.e., from low to high, from high to low) would help to convey more information.

• Page 5, line 198, why do authors choose 50 miles as the threshold? Is there any reference for it?

• There are some typos in the manuscript. For example,

o Page 4, line 143, it should be “National Park Service” instead of “National Park Servive”

o page 8, line 301, it should be “grouping” instead of “groupping”.

o Page 8, line 312, it should be “fitted” instead of “fit”

Reviewer #2: This study presents an attempt to utilize human mobility data (celluar device location here) to estimate the visits to public lands. While I think the paper is of certain interests, the paper needs certain revision to be publishable. I especially have concerns about the writing and references of the study.

1. The abstract needs to substantial revision. The authors should first give a sentence of the broader context of the background, rather than directly jumping into what you did.

2. Besides, in the abstract, too many sentences started from “We”, I suggest revise these sentences to make them more academically professional and diverse. I also suggest the authors do not include so many findings, but highlight the important points or implications out of the findings.

3. “However, this new source …. collected by more traditional methods.” Not sure why this is included in the abstract. Also, maybe the authors can consider talking about the meaning/value of your study in broader context, to other fields or real-world practice in general.

4. More importantly, I suggest the authors should carefully polish the motivations of this study. Especially, the authors can discuss the gaps left by previous works and how the study fills such gap. Has previous works done similar things? What are the important issues yet not been addressed? These are the important storyline to be told, in Abstract and especially in Introduction.

5. I suggest including a table to show the sources and statistics of the data used in this study.

6. In section 3.1, I suggest give a concise and clear implications out of your findings. The current section looks like a piles of the results without interpretation.

7. I think there lacks a discussion of the limitations/potential directions for future works.

8. There are certain important relevant works missed by the authors: “Categorisation of cultural tourism attractions by tourist preference using location-based social network data: The case of Central, Hong Kong” (which talks about human mobility for tourism categorization and profiling), “A review of human mobility research based on big data and its implication for smart city development” (an important review for human mobility application for smart cities), “Smart tourism destinations: An extended conception of smart cities focusing on human mobility” (another important study at the intersection of human mobility&tourism)

6. PLOS authors have the option to publish the peer review history of their article (what does this mean?). If published, this will include your full peer review and any attached files.

Reviewer #1: No

Reviewer #2: No

---

## [Author Response · Author response to Decision Letter 0]

20 Jul 2023

Please see attached "Responses To Reviewers" file for responses to reviewers.

Reviewer #1

Thank you for the opportunity to review your manuscript. The study explored the use of commercially available cellular data for estimating visitation to public lands by comparing it with NPS visitor use statistics. It is a well-written, needed, and useful study of the use of cell data. The manuscript effectively summarizes the methods and results. Here are some recommendations:

Reponses: 

First and foremost, we greatly appreciate your time in providing us valuable feedback that helped us enhance the clarity and improve the quality of the manuscript. We have addressed the following comments accordingly and detailed our responses below. 

• In the introduction section, the manuscript needs to emphasize the unique contribution of this study. On page 3, the authors listed many existing studies that found cell data and on-site counts were comparable. What are the research gaps? Why is this study needed? What are the theoretical, methodological, or empirical contributions of this study?

Reponses:

Thank you very much for pointing this out. We have revised the Introduction section accordingly.

We added sentences to clarify the research gap in Line #109 – 111, “However, most of the existing studies using cell data to estimate visitation are applied in one site or with relatively similar geographic characteristics. Applying cell data across a large geographic area likely with great variations in settings (e.g., urbanicity, recognition) remains under explored.”

We also explained our study aims and contributions to the research area in Line #120 – 121, “To understand the feasibility of cell data for quantifying visitation information and to address the research gap on the application of cell data across a large spatial scale,…”

• The study explored the role of park type, park setting, and porousness in explaining differences between the cell data and NPS recreation visit estimates. There might be some other important factors. For example, geographic characteristics might influence the quality of cell data. The proportion of different modes to access park units, such as private vehicles, biking, walking, bus, and boat, might affect the NPS counts. If these data are not available, these issues need to be discussed as limitations.

Reponses:

We agree but the NPS does not provide estimates for how many people arrive via the various transportation modes (e.g., private vehicles, biking, walking, bus, etc.) and there is no way of extracting that information from the data provided. Transportation mode, however, is somewhat reflected by primary counting methods employed by parks and we did explore the potential differences in the relationships between NPS and cell data counts by the primary on-the-ground counting methods used in each park. We found that it is difficult to clearly discern the relationships by on-the-ground counting method, since most of the parks use a combination of methods, such as traffic counter + bus count + estimation based on other sites + on-site observation, etc. NPS does not report how many people are counted by each one of their counting methods. Thus, we came up with other attributes (distance to population center, recognition from non-local communities, porousness) that can more clearly classify parks into different groups in this study.

We added the following sentences below in Line #372 – 381 to address this limitation.

“We investigated the potential variations in the relationships between NPS and cell data counts by park attributes we felt were important and for which we could compile data. We were not able to find data for all attributes we would have liked to examine. For example, we would have liked to include park access transportation mode (e.g., walking, private vehicle, bus, biking, horseback, ferry, etc.), but the NPS does not collect these data. Park access mode is somewhat reflected by the NPS counting methods. However, we found that most of the parks use a combination of counting methods and do not provide visitor counts by each counting method. On-the-ground counting data collections are usually complex (e.g., a combination of traffic counter with various adjustments for person-per-vehicle at different entrances and/or seasons, cars in the camping sites, and assuming a constant number per bus).”

• On page 10, the manuscript states, “the cell data may be appropriate for trends analysis.” Although this thought is interesting and valuable, more data and evidence are needed to support this argument.

Reponses:

We agree. We decided the discussion was beyond the scope of the paper and removed this statement. It will likely be addressed in future work. 

• Page 10, the manuscript states, “It may also be useful to capture use in areas of a park where on-the-ground counts are impractical.” More explanations are needed to describe the situations when on-the-ground counts are impractical.

Reponses:

We changed this sentence to read “It may also be useful to capture use in areas of a park where on-the-ground counts are impractical, such as on remote Bureau of Land Management lands where counts are not conducted, or on city parks with many entrances and lack of resources for surveys.” in Line #428 – 430.

• Figure 2, in each category, parks are ordered alphabetically. Reordering the parks based on the proportion of local visitors (i.e., from low to high, from high to low) would help to convey more information.

Reponses:

We reordered the y-axis by the percentage of visitor coming within 50 miles. 

• Page 5, line 198, why do authors choose 50 miles as the threshold? Is there any reference for it?

Reponses:

We referenced the definition of “local visitors” used in the U.S. Forest Service National Visitor Use Monitoring program. Please see page 9 in https://www.fs.usda.gov/sites/default/files/2021-National-Visitor-Use-Monitoring-Summary-Report.pdf

We added the information and citation after the original sentence, “We calculated Euclidean distance from the centroid of the POI (i.e., parks) to home CBGs and defined any pair of distance within 50 miles as local commuting areas”, in Line #214 – 215, “…based on the definition of local visitor used in the U.S. Forest Service National Visitor Use Monitoring program (49).”

Added references:

49. U.S. Forest Service. U.S. Forest Service National Visitor Use Monitoring Survey Results National Summary Report: Data collected FY 2017 through FY 2021. Available at https://www.fs.usda.gov/sites/default/files/2021-National-Visitor-Use-Monitoring-Summary-Report.pdf. 2021.

• There are some typos in the manuscript. For example,

o Page 4, line 143, it should be “National Park Service” instead of “National Park Servive”

o page 8, line 301, it should be “grouping” instead of “groupping”.

o Page 8, line 312, it should be “fitted” instead of “fit”

Reponses:

We fixed these edits and typos.

Reviewer #2

This study presents an attempt to utilize human mobility data (celluar device location here) to estimate the visits to public lands. While I think the paper is of certain interests, the paper needs certain revision to be publishable. I especially have concerns about the writing and references of the study.

Reponses:

First and foremost, we greatly appreciate your time in providing us valuable feedback and references that help us enhance the clarity and improve the quality of the manuscript. We revised the abstract substantially and added the references you provided. We also detailed our responses to each comment below.

1. The abstract needs to substantial revision. The authors should first give a sentence of the broader context of the background, rather than directly jumping into what you did.

Reponses:

We substantially edited the abstract to address this and other points made.

2. Besides, in the abstract, too many sentences started from “We”, I suggest revise these sentences to make them more academically professional and diverse. I also suggest the authors do not include so many findings, but highlight the important points or implications out of the findings.

Reponses:

We changed up the way we started sentences to diversify the language in the abstract. We limited the findings and highlighted the main contribution. 

3. “However, this new source …. collected by more traditional methods.” Not sure why this is included in the abstract. Also, maybe the authors can consider talking about the meaning/value of your study in broader context, to other fields or real-world practice in general.

Reponses:

We re-wrote the abstract to address these concerns. 

4. More importantly, I suggest the authors should carefully polish the motivations of this study. Especially, the authors can discuss the gaps left by previous works and how the study fills such gap. Has previous works done similar things? What are the important issues yet not been addressed? These are the important storyline to be told, in Abstract and especially in Introduction.

Reponses:

Pointing out specific previous research and the gaps we left to the introduction and literature review sections. We edited the abstract to highlight our contribution in general. 

5. I suggest including a table to show the sources and statistics of the data used in this study.

Reponses:

Thank you very much for the suggestion. We summarized the data sources and provided summary statistics for the total visitation and visitation by month from both data sources in Supplemental Table S1. 

We also added the text to refer to this Supplemental Table in Line #183 – 184, “A summary of the data sources and statistics of total visitation used in this analysis are described in S1 Table.”

6. In section 3.1, I suggest give a concise and clear implications out of your findings. The current section looks like a piles of the results without interpretation.

Reponses:

We revised the section to provide a short summary of observation.

7. I think there lacks a discussion of the limitations/potential directions for future works.

Reponses:

Our discussion section addresses many limitations (differences across parks, not being able to discern whether the cell data or NPS counts are inaccurate, stability of the data provider). We then discuss future directions (representativeness, stability over time, building a prediction model). We edited the discussion section based on this and other comments.

8. There are certain important relevant works missed by the authors: “Categorisation of cultural tourism attractions by tourist preference using location-based social network data: The case of Central, Hong Kong” (which talks about human mobility for tourism categorization and profiling), “A review of human mobility research based on big data and its implication for smart city development” (an important review for human mobility application for smart cities), “Smart tourism destinations: An extended conception of smart cities focusing on human mobility” (another important study at the intersection of human mobility&tourism)

Reponses:

Thank you very much for providing these important references. 

We added Liu et al., 2022 as one of the citations for the use of social media on spatial patterns and activities in Line #81, and added the following sentences with Wang et al. (2021) and Lamsfus et al. (2015) as the citations.

“Like many other sources of big data (e.g., vehicle GPS, metro card, bank card), cell data can not only help understand human mobility patterns but also social, cultural, and economic values for better land management and city development (42, 43)” in Line #106 – 108.

Added references:

29. Liu Z, Wang A, Weber K, Chan EHW, Shi W. Categorisation of cultural tourism attractions by tourist preference using location-based social network data: The case of Central, Hong Kong. Tourism Management. 2022;90:104488.

42. Lamsfus C, Martin D, Alzua-Sorzabal A, Torres-Manzanera E, editors. Smart tourism destinations: An extended conception of smart cities focusing on human mobility. Information and Communication Technologies in Tourism 2015: Proceedings of the International Conference in Lugano, Switzerland, February 3-6, 2015; 2015: Springer.

43. Wang A, Zhang A, Chan EHW, Shi W, Zhou X, Liu Z. A Review of Human Mobility Research Based on Big Data and Its Implication for Smart City Development. ISPRS International Journal of Geo-Information. 2021;10(1):13.

---

## [Decision Letter · Decision Letter 1]

31 Jul 2023

Using cellular device location data to estimate visitation to public lands: Comparing device location data to U.S. National Park Service’s visitor use statistics

PONE-D-23-15317R1

Dear Dr. Tsai,

We’re pleased to inform you that your manuscript has been judged scientifically suitable for publication and will be formally accepted for publication once it meets all outstanding technical requirements.

Kind regards,

Zihao Zhang

Academic Editor

PLOS ONE

Additional Editor Comments (optional):

Reviewers' comments:

Reviewer's Responses to Questions

**Comments to the Author**

1. If the authors have adequately addressed your comments raised in a previous round of review and you feel that this manuscript is now acceptable for publication, you may indicate that here to bypass the “Comments to the Author” section, enter your conflict of interest statement in the “Confidential to Editor” section, and submit your "Accept" recommendation.

Reviewer #1: All comments have been addressed

Reviewer #2: All comments have been addressed

2. Is the manuscript technically sound, and do the data support the conclusions?

Reviewer #1: Yes

Reviewer #2: Yes

3. Has the statistical analysis been performed appropriately and rigorously? 

Reviewer #1: Yes

Reviewer #2: Yes

4. Have the authors made all data underlying the findings in their manuscript fully available?

Reviewer #1: No

Reviewer #2: Yes

5. Is the manuscript presented in an intelligible fashion and written in standard English?

Reviewer #1: Yes

Reviewer #2: Yes

6. Review Comments to the Author

Reviewer #1: Thank you for addressing my questions and comments. Great improvements have been made in the abstract, introduction, limitation, and reference sections.

Reviewer #2: The authors have addressed my concerned. The manuscript has been much improved and I recommend acceptance.

7. PLOS authors have the option to publish the peer review history of their article (what does this mean?). If published, this will include your full peer review and any attached files.

Reviewer #1: No

Reviewer #2: No

---

## [Editor Report · Acceptance letter]

3 Aug 2023

PONE-D-23-15317R1 

Using cellular device location data to estimate visitation to public lands: Comparing device location data to U.S. National Park Service’s visitor use statistics 

Dear Dr. Tsai:

I'm pleased to inform you that your manuscript has been deemed suitable for publication in PLOS ONE. Congratulations! Your manuscript is now with our production department. 

Kind regards, 

on behalf of

Dr. Zihao Zhang 

Academic Editor

PLOS ONE